# AI-determined similarity increases likability and trustworthiness of human voices

**Oliver Jaggy**[1]*, **Stephan Schwan**[1], **Hauke S. Meyerhoff**[2]

**1** Leibniz-Institut für Wissensmedien, Tübingen, Baden-Württemberg, Germany, **2** University of Erfurt, Erfurt, Thüringen, Germany

☯ These authors contributed equally to this work.

* o.jaggy@iwm-tuebingen.de

## Abstract

Modern artificial intelligence (AI) technology is capable of generating human sounding voices that could be used to deceive recipients in various contexts (e.g., deep fakes). Given the increasing accessibility of this technology and its potential societal implications, the present study conducted online experiments using original data to investigate the validity of AI-based voice similarity measures and their impact on trustworthiness and likability. Correlation analyses revealed that voiceprints – numerical representations of voices derived from a speaker verification system – can be used to approximate human (dis)similarity ratings. With regard to cognitive evaluations, we observed that voices similar to one's own voice increased trustworthiness and likability, whereas average voices did not elicit such effects. These findings suggest a preference for self-similar voices and underscore the risks associated with the misuse of AI in generating persuasive artificial voices from brief voice samples.

## Introduction

Artificial intelligence (AI) has become integral to modern life and is revolutionizing how people interact with technology and process information. From autonomous vehicles to personalized recommendation systems, AI's ability to analyze and replicate human-like behaviors profoundly impacts all industries. One particularly relevant application is in the field of speech technology, in which AI systems not only recognize and synthesize speech but also simulate individual voice characteristics. This capability opens new avenues for personalized interactions, such as matching voice assistants to a user's voice profile or augmenting that profile, illustrating the interplay between technology, identity, and human perception.

The human voice remains a remarkable signature of individuality, transcending mere communication to embed rich layers of information about the speaker. Beyond the conveyance of words, each voice carries the unique timbre, tone, and other acoustic information, that hint at the speaker's identity. Like fingerprints, the human voice can be used to distinguish individuals from one another with a high degree of accuracy [1,2] and gives insights into the speaker's emotions and physical attributes. Speech data can be used, for example, to recognize stress [3],

**Data availability statement:** All anonymized data files are available from the

figshare database https://doi.org/10.6084/m9.figshare.21022081.v2

**Funding:** The author(s) received no specific funding for this work.

**Competing interests:** The authors have declared that no competing interests exist.

emotions [4–7], the level of interest [8], age and sex [9,10], and personality traits [11,12] – for a review on speech analysis for health, see [13].

Voice assistants such as Alexa or Siri attempt to mimic human voice in terms of pleasant and recognizably individualized speech characteristics. So far, most voice assistants implement only one synthetic voice and thus follow an approach in which one voice fits all users [14]. However, voice assistants may also compute the voiceprint (see below) of the customer and utilize this information to modify the synthetic voice to make it similar to the customer's voice.

Therefore, the question arises how listeners evaluate voices similar to the listeners' own voices and whether they prefer average voices compared to more distinct voices. The present paper addresses this question in five experiments. As a first step, we show that similarity judgements of two voices by AI-based speaker recognition systems and human listeners significantly correspond (Experiments 1 and 2), which is a necessary precondition for AI-based cloning of individual voices. As a second step, we show that this correspondence also holds if one of the voices is the listener's own voice (Experiment 3). As a third step, we showed that average voices are not preferred over distinct voices (Experiment 4). We finally demonstrate that listeners judge voices similar to their own voice (according to the AI-based speaker recognition system) to be more likable and trustworthy then dissimilar voices (Experiment 5).

## Characterizing individual human voices through AI-based d-vectors

The complexity of human speech poses a significant challenge: how can one distill and encode these sophisticated vocal characteristics into a form that captures the essence of individual identity? Modern speaker recognition systems use d-vectors, or similar kinds of speaker embeddings (such as x-vectors, r-vectors, or ECAPA-TDNN), derived from a deep neural net [15,16]. While there are only minor differences in performance among these speaker embeddings [17], d-vectors have the advantage that the speaker encoder that generates the embeddings is a lightweight model, widely used in the open-source community, and relatively easy to implement.

Starting point are short audio samples of a human speakers. The audio samples are non-linearly transformed on the frequency scale to emphasize distances between frequencies for which the human ear is most sensitive. Next, these transformations, called mel-spectrograms, are used to train deep neural networks. D-vectors, then, are the averaged activations of the final hidden layer of a deep neural network that is trained on a speaker verification or identification task. As a result, they are abstract representations of audio called "voiceprints", which contain compressed information about the audio signal's unique characteristics, such as timbre and tone, in a multidimensional space.

However, such voiceprints may not only be used for speaker identification. Instead, deep learning methods [18,19], enables software to clone the voice of a real person. Cloning a voice traditionally involves training a Text-to-Speech (TTS) system using audio samples from the target individual. However, this requires a large number of audio samples from the individual (often unavailable) and the training of an entire TTS system, which is both time-intensive and computationally demanding. Consequently, cloning someone's voice was most of the time either impossible or too prohibitively expensive. Yet, providing voiceprints as additional information when training a TTS system makes it possible to clone a voice with only a few seconds of audio material and without the need to train a new system [20–22]. Even if the results are not yet as convincing as previously used techniques, the essential prerequisites have been met to convert any given text into speech and predetermine the used voice by providing a voiceprint.

## Comparing human to d-vector based voice similarity judgments

Yet, there is little research on how voiceprints are related to human perception. Since voiceprints are new in the field, we needed to establish their validity for (human) similarity judgments, which is a prerequisite to study the cognitive consequences of voice similarity thereafter. Beside research on performance differences between human and speaker recognition systems [23], to the best of our knowledge, there is only one study that has investigated the relationship between voice similarity estimates by humans and an automatic speaker recognition system [24]. The study by [24] showed a positive relationship between participants' similarity judgments and comparison scores from a speaker recognition system [25]. In contrast to this study, we are interested in voiceprints derived from a speaker recognition system based on d-vectors, which are derived by training a deep neural net [26] and can be used to clone a voice. Although our study does not employ cloned voices, the application of d-vectors for generating speech that resembles a target speaker's voice makes them the ideal candidate for examining similarity effects.

## The significance of likeability and trustworthiness in social interactions

Likeability and trustworthiness are foundational attributes that significantly influence social interactions and relationships. Research has demonstrated that individuals judged more likable by others are more persuasive, often receiving preferential treatment and social support [27–30], and that similar others are also perceived as more likeable [31]. Similarly, trustworthiness is central in fostering long-term (business) relationships and ensuring effective collaboration, as it mitigates uncertainty, reduces the perceived risk in interactions and increases predictability [32–36]. From an evolutionary perspective, trustworthiness likely signals an individual's reliability and cooperative intent, which are essential for fostering social cohesion and reciprocal behaviors within groups. Similarly, likeability facilitates social bonding by eliciting positive affect and reducing interpersonal tension, enhancing collaboration and mutual support. Thus, since these constructs are integral to social evaluation processes, using likability and trustworthiness as dependent variables is critical to understanding the impact of voice similarity.

## Voice typicality and its influence on trustworthiness and likability

Previous research in other perceptual domains has consistently demonstrated a *beauty-in-averageness* effect, where average or prototypical faces and objects are perceived as more attractive than those that deviate from the norm [37–39]. This phenomenon extends beyond mere aesthetic preference, reflecting a broader cognitive tendency to favor typical over unusual stimuli, which may be rooted in the ease of processing more familiar or expected patterns [40]. Additionally, familiarity has been shown to enhance social evaluations, such as perceived trustworthiness, particularly in the context of faces [41]. These findings suggest that perceptual and cognitive processes prioritize typicality and familiarity, potentially because they signal safety, reliability, or group affiliation.

Building on this framework, Experiment 4 sought to explore whether a similar effect is observable in the auditory domain, specifically for voices. In this context, typicality was operationalized as the mean cosine similarity between a given speaker's voiceprint – a numerical representation of their vocal characteristics – and the voiceprints of all other speakers in our dataset. By examining whether voices with higher typicality are associated with greater trustworthiness and likability, we aimed to extend the beauty-in-averageness principle to auditory stimuli.

## Similarity attraction and the possible effects of voices resembling listeners' own voices

Cloned voices are an essential component of deep fakes, which are primarily used for entertainment purposes, such as showing Elon Musk performing a belly dance or Barack Obama mocking Donald Trump. However, there are also malicious use cases [42], and deep fakes have been used to spread fake news and propaganda [43]. However, there are also more subtle possibilities to use manipulated audio, particularly in the field of voice assistants, which could have significant effects on users of voice assistant systems. According to the *similarity attraction hypothesis* [44,45], people like other people more if they behave, appear or think similarly to them – for a meta-analysis, see [46]. A possible explanation for similarity attraction is linked to a phenomenon called *implicit egotism*: People tend to evaluate themselves positively, and if they associate other people with themselves, the positive self-evaluation may influence their evaluation [47–49]. Building on this concept, it has been proposed that similarity influences attraction by shaping the perceived valence and significance of inferred traits [50–52]. Specifically, individuals may derive positive or negative evaluations of others based on shared or divergent attitudes, personality traits, or other attributes. According to [51], similar attitudes do not directly lead to attraction but foster expectations of additional positive qualities in the similar individual, driven by the individual's own favorable self-assessment [50,53]. Moreover, the inclination to interpret superficial similarities as indicative of deeper shared traits can result in an overestimated sense of alignment. For instance, individuals might presume that an advisor who shares surface-level characteristics also holds similar preferences, thereby perceiving their advice as more applicable or insightful [54]. However, the effects of similarity may also stem from humans' inherently social nature. *Social Identity Theory* [55] suggests that individuals exhibit a preference for and more positive behaviors toward those they perceive as members of their own group.

Biological explanations further contribute to our understanding of similarity effects. In the context of social networks, this mechanism is often referred to as homophily, which describes the tendency to form connections with similar others [56]. From an evolutionary perspective, it has been argued that altruistic behaviors typically come at a cost, except when directed toward genetically related individuals [57,58]. Since genetic similarity often correlates with phenotypic resemblance, evolutionary pressures may have favored prosocial behaviors toward those perceived as similar, enhancing cooperation and cohesion within social groups.

However, similarity effects do not only occur between human agents. Research on human-machine communication has shown that humans exhibit social responses to computers just as they do to humans. Consequently, similarity attraction also might arise in human-computer interactions involving artificial voices [59,60]. Indeed, general (i.e. non-adaptive) alignments of acoustic-prosodic features, such as speech rate, intensity, pitch, volume, and prosody, can lead to a similarity attraction towards synthetic voices [61], which can positively influence learning [62], engagement [63], and enjoyment [64].

## Based on the above considerations, we investigated

- Whether the cosine similarity derived from the trained neural network correlates with human similarity judgments (Exp 1-3).

- Whether speakers with prototypical voices are judged as more likeable and trustworthy (Exp 4).

- Whether speakers with similar voiceprints to the corresponding participants are perceived as more likable and trustworthy (Exp 5).

## The relation between AI and human similarity judgments

In the first Experiment, we investigated the validity of the cosine similarity as a measure of perceived voice similarity by probing whether the cosine similarity of two voiceprints predicts human similarity judgments.

### Method

**Ethics statement.** The studies reported were approved by the ethics committee of the Leibniz-Institut für Wissensmedien, Tübingen (approval number LEK 2020/061 and LEK 2021/123). All participants provided written informed consent through the online platform qualtrics.com, and all experiments included in this study were preregistered (Exp. 1: https://osf.io/kxwsv; Exp. 2: https://osf.io/8c7xw; Exp. 3: https://osf.io/yt3b7; Exp. 4: https://osf.io/q59da; Exp. 5: https://osf.io/cv5g9).

**Encoder and data.** For the first as well as the following Experiments, we used an open-source encoder [65] based on research conducted by [22,66,67]. In contrast to the described model in [65], our model consists of three recurrent neural networks (RNN) of the long short-term memory type(LSTM layers) with 768 nodes, followed by a fully connected projection layer with 256 nodes and a tanh activation function.

The encoder is trained on a speaker verification task in which it learns to embed utterances from the same speaker close together in the embedding space and utterances from different speakers farther away. This increases intraspeaker variation and enhances interspeaker discrimination. For each utterance, a 256-dimensional feature vector is created, where each feature can encode certain voice features. These voice features are characteristic for the speaker and could be understood as a numerical representation of the voice, a voiceprint. The similarity of two voices can be compared by calculating the cosine similarity of two feature vectors, yielding values ranging from -1 to 1.

To train the network, we used the German subset of the Common-Voice dataset (https://github.com/mozilla/common-voice), Distant-Speech (https://www.inf.uni-hamburg.de/en/inst/ab/lt/resources/data/acoustic-models.html), LibriVoxDeEn [68], and the German subset of the VoxForge dataset (http://www.voxforge.org/home). These datasets contain linguistically diverse material. The linguistic content ranges from simple phrases to complex sentences, covering a broad spectrum of phonetic, lexical, and syntactic structures in German. The combined datasets consist of approximately 1,000 hours of spoken audiobooks and Wikipedia articles read aloud by about 10,000 non-professional speakers. Audio files not already cut at sentence boundaries were cut at the appropriate points.

**Participants.** We chose a sample size of 100 participants for our first experiment as a practical starting point. This sample size provided a balance between feasibility and statistical power, allowing us to evaluate the relationship between cosine similarity and human similarity judgments while accounting for individual variability. Therefore, we recruited 50 male and 50 female German participants via prolific (https://www.prolific.com), which was the recruitment platform used for all experiments. Basic demographic information was collected via Qualtrics (https://qualtrics.com) in all experiments.

Six of the participants were excluded because they failed in more than one control trial. The mean age was $M = 32.01$ ($SD = 11.26$). Forty-seven of the 94 participants were female, one diverse, and three refused to answer. Recruitment occurred from March 3, 2021, to March 5, 2021. Participants received £3.45 for their participation in the study.

**Materials, stimuli and procedure.** Since our dataset included more male than female speakers (approximate ratio of 3:1), and because this was our first experiment using this type of data, we aimed to achieve a wide range of cosine similarity values with high granularity.

We used only male speakers in this study to simplify the experimental design and ensure consistent conditions.

For each male speaker in our dataset, we calculated the cosine similarity of the voice embedding with each other speaker in the dataset. Since those cosine similarities are approximately normally distributed, randomly drawing from these pairs would result in too few examples from the edge categories. Therefore, we subdivided the cosine values into ten categories, using the lowest and highest cosine value between speaker pairs as reference points with equal cosine value differences between the breakpoints. We subsequently drew speaker pairs based on the categories, which should ensure an even distribution of cosine values and, therefore, the greatest possible variance in the stimulus material. For each drawn speaker, we randomly picked one audio sample from our dataset, trimmed it to a maximum length of 5 s, and normalized the volume. We drew 50 sets of 100 male speaker pairs and presented each set to one female and one male participant in a random order.

Since our experiments were conducted online on pavlovia.org (https://pavlovia.org/) using PsychoPy [69], we checked whether participants had a working audio setup at the beginning of the experiment: We presented a short text in which we informed them to count sinus tones. After presenting four sinus tones with an inter-stimulus interval of 1s, participants should indicate on a slider (ticks on 0,1, 2,3,4 and 5) how many sinus tones they were hearing. If they failed the test, the experiment was concluded.If they passed the detection task, three introductory trials were presented that had the same structure as the regular trials; audio recordings from two different male speakers were presented sequentially, with an inter-stimulus interval of 1s. While audio was played, a headphone icon was depicted. After hearing both voices once, the participants were asked to rate the dissimilarity by adjusting the slider on an unmarked continuous rating scale (range: little dissimilarity - great dissimilarity). They were allowed to take as much time as needed for this rating, with no imposed time constraints. While piloting our study, we found it much more challenging to rate the similarity than the dissimilarity. Accordingly, we asked the participants to rate the dissimilarity rather than the similarity and inverted the response afterward. Participants could skip a rating but were informed that they should only choose this option if they couldn't hear one of the samples properly. The participants who skipped more than ten trials were excluded. To check the participants' attention, we presented every 30th trial two different audio samples from the same speaker. The participants who rated the dissimilarity higher than 0.2 in more than one of the three control trials were excluded.

## Results

To analyze the data in this study, we used the software R [70], the R package lme4 [71], and the R package MuMIn [72]. We used mixed models with participants as random effects, the raw cosine values as the independent variable, and the inverted slider responses as the dependent variable (R code for data processing is publicly available, see below). To test whether encoder ratings can predict how similarly humans judge different voices, we compared an intercept-only model, a linear model, and a quadratic model. We included a quadratic model because human judgments, particularly those based on perceptual features like voice similarity, often show non-linear trends [73]. This approach accounts for potential non-linear relationships between the cosine similarity of voice embeddings and human similarity judgments. For instance, individuals may perceive two voices as more similar up to a certain point, but after that, additional increases in cosine similarity might not yield proportional increases in perceived similarity. This suggests diminishing or varying returns on perceived similarity as cosine similarity increases.

Since previous research found evidence for a own-gender bias in the ability of voice identification [74] and gender differences in voice processing [75,76], we included participants' gender as an additional factor. Weighting using Akaike information criterion (AIC) scores (see Table 1) showed a clear quadratic relationship between the calculated cosine similarity of the encoder and participants' rating (intercept: 0.36, *95% CI* [0.34, 0.38], $t(102.3) = 33.05$, $p < .001$; cosine: 0.10, *95% CI* [0.07, 0.13], $t(9215) = 7.68$, $p < .001$; cosine$^2$: 0.47, *95% CI* [0.42, 0.52], $t(9215) = 18.51$, $p < .001$). The median of the individual Spearman Correlations between the encoder's cosine similaritiy values and participants' similarity ratings was *Mdn* $r_s = 0.37$ ($Q_1 = 0.28$, $Q_3 = 0.43$), indicating a moderate relationship. The model explained 27% of the variance ($R_c^2 = 0.27$). Numerically, female participants rated the similarity slightly higher; however, the inclusion of gender as an additional factor is not justified given the AIC values.

The relationship between cosine values and the participants' similarity ratings seems stronger for higher cosine similarity values. The quadratic relationship indicates that the cosine values derived from the deep neural network are associated with human similarity judgments and highlight a stronger association at extremer cosine similarity values. Participants skipped, on average, $M = 0.86$ trials (SD = 1.16) and needed, on average, *Mdn* = 28.98 minutes to complete the experiment.

Given the large variance observed among participants' similarity ratings and the use of a vast amount of different stimulus pairs, we performed additional analysis with aggregated data to gain more insights into the relationship between cosine values and similarity judgments. Rather than employing the encoder's raw cosine values for each similarity judgment, we utilized the predefined similarity categories used in the sampling process as predictors. The response variable was the mean similarity judgment corresponding to each category. Since the above analysis revealed a quadratic relationship, we compared a linear model with a quadratic regression model using an analysis of variance (ANOVA). The results strongly favored the quadratic model over the linear model, $F(1,7) = 240.98$, $p < .001$. The analysis of the quadratic model itself revealed a significant quadratic relationship between the similarity category and the mean similarity rating, $F(2,7) = 670.5$, $p < .001$ (intercept: 0.37, *95% CI* [0.36, 0.39], $t(7) = 50.51$, $p < .001$; category -0.02, *95% CI* [-0.03, -0.01], $t(7) = -5.97$, $p < .001$, category2: 0.006, *95% CI* [0.005, 0.007], $t(7) = 15.52$, $p < .001$). The model explained most of the variance in the mean similarity rating, $R^2 = 0.995$ (Fig 1). Even though mixed effects models account for random variation and lead to shrinkage, the analysis with more aggregated data further reduces variation and leads to a more pronounced relation.

While the association between similarity values and similarity judgments appears modest in the analysis with mixed models, the analysis with aggregated data suggests that the relationship warrants attention. This finding is noteworthy given the potential limitations of using stimuli derived from open-source datasets. Factors such as varying audio quality, speakers' articulation proficiency, and the semantic content of audio clips could have influenced evaluations. At the same time, the diversity of the stimuli may have contributed to the ecological validity of incidental similarity evaluations.

**Table 1. Model selection table for Experiment 1.**

| Model name | Degrees of freedom | Log-likelihood | AIC | ΔAIC | Weight |
|---|---|---|---|---|---|
| Quadratic model | 5 | −354.73 | 719.5 | – | 0.911 |
| Quadratic model + gender | 6 | −356.04 | 724.1 | 4.64 | 0.089 |
| Linear model | 4 | −520.26 | 1048.5 | 329.07 | 0 |
| Linear model + gender | 5 | −521.59 | 1053.2 | 333.74 | 0 |
| Intercept-only model | 3 | −1189.42 | 2384.8 | 1665.39 | 0 |

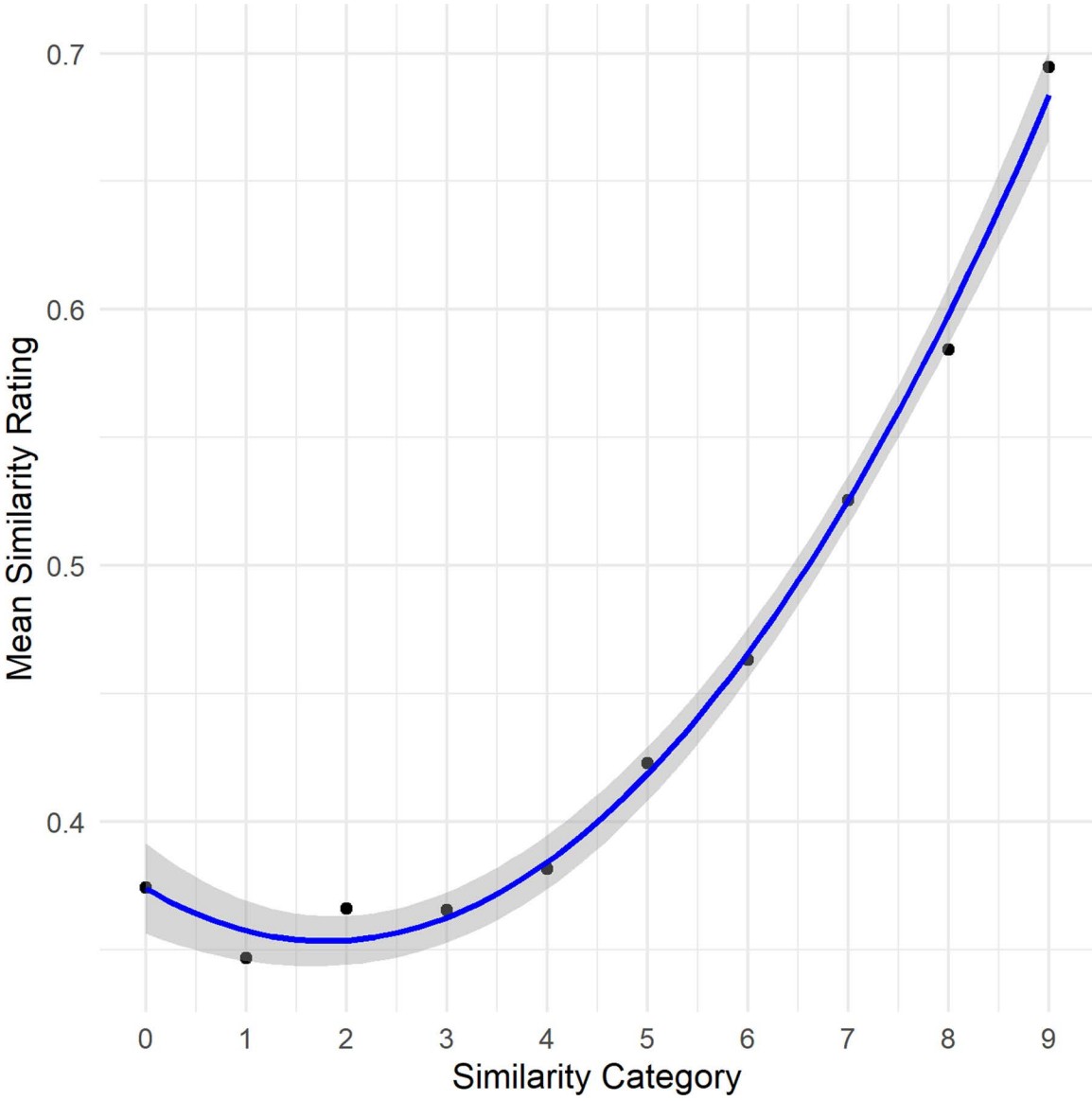

**Fig 1. Illustration of the results of Experiment 1.** Depicted is the quadratic relationship between cosine similarity categories and the participants' mean similarity ratings as well as the the 95% confidence interval of the regression line.

It is also important to consider that assessing the similarity of two voices based on just 5 seconds of random samples is inherently challenging. The quadratic relationship observed in the data indicates that these challenges were particularly pronounced when participants evaluated speaker pairs with moderate to low cosine similarity values. These complexities and their potential implications for interpreting the results will be explored further in the General discussion.

Our findings gain context when compared with those of MOSNet [77]. MOSNet demonstrated a capacity to predict human-perceived similarity judgments, more precisely termed identity ratings, with Spearman Rank Correlation coefficients ranging between 0.292 and 0.455. The derived median correlation coefficient of 0.37 in our experiment aligns with the midpoint of MOSNet's observed correlations but for raw values on a similarity judgment

– which we consider more valuable for research on the influence of voice similarity on cognitive processes.

Overall, the results confirm the validity of cosine similarity as a measure of perceived voice similarity. The quadratic relationship suggests that participants were disproportionately sensitive to very dissimilar and very similar voices but less capable of differentiating at intermediate similarity levels. This may reflect a natural limit in human voice discrimination abilities, particularly for voices that are neither too distinct nor too similar. These findings support the utility of AI-generated cosine similarity for approximating human voice similarity judgments.

## The reliability of human similarity judgments

In order to interpret the magnitude of the correlation between raw cosine values and similarity judgments observed in Experiment 1, we needed to assess the reliability of human similarity judgments, which limits the maximum of observable correlations [78].

### Method

**Participants.** G*Power [79] was used to calculate the necessary sample size for the Correlation: Bivariate normal model test as an approximation for the non-parametric Spearman rank correlation test that was used to calculate the test-retest reliability. The analysis aimed to detect a medium to large effect size with an alpha = 0.05 and a power = 0.80. The power analysis revealed a minimum sample size of 46 participants. We therefore recruited 50 new participants via prolific. Five were excluded because they failed in more than one control trial. Eighteen of the remaining 45 participants were female, two did not indicate their sex. The mean age was $M = 30.31$ ($SD = 10.91$). Recruitment took place on April 19, 2021. Participants received £4.36 for their participation in the study.

**Materials, stimuli and procedure.** Besides some minor changes, the material and procedure were identical to Experiment 1. In contrast to Experiment 1, we sampled 50 instead of 100 speaker pairs and only used one set of speakers for all participants. After each of the 50 speaker pairs was presented once, participants were asked to rate their similarity again. The order of the speaker pairs was altered in the second part of the experiment but was the same for all participants. We aimed for this uniformity of the experimental conditions to avoid variance from individual randomizations of the trials order since our approach mainly focused on correlations, which require reliable estimates for person parameters rather than experimental conditions (for which randomization would be necessary).

### Results

As in Experiment 1, we observed a correlation between cosine similarity and similarity judgments. Using AIC values from Table 2, we identified a quadratic relationship between cosine similarity (generated by the encoder) and participants' ratings. The model explained 19% of the variance in similarity judgments (intercept: 0.43, *95% CI*: 0. 40 to 0. 46, $t(47.05) = 25.75$, $p < .001$; cosine: 0. 09, *95% CI* [0. 05, 0.13], $t(4249) = 4.55$, $p < .001$; $cosine^2$: 0. 24, *95% CI* [0. 16, 0. 31], $t(4249) = 5.97$, $p < .001$, $R_c^2 = 0.19$). The median of the individual Spearman

**Table 2. Model selection table for Experiment 2.**

| Model name | Degrees of freedom | Log-likelihood | AIC | ΔAIC | Weight |
|---|---|---|---|---|---|
| Quadratic model | 5 | −332.97 | 675.9 | – | 1 |
| Linear model | 4 | −348.40 | 704.8 | 28.85 | 0 |
| Intercept-only model | 3 | −459.12 | 924.2 | 248.30 | 0 |

Correlation between cosine similarities and similarity ratings was $Mdn\ r_s = 0.23$ ($Q_1 = 0.19$, $Q_3 = 0.26$). Therefore, we were able to replicate our results of Experiment 1, which showed a quadratic relation between cosine similarities and similarity judgments. The relationship, however, was slightly less pronounced than in Experiment 1. This may be explained by the limited stimulus material required to measure the reliability scores as well as the reduced number of trials to keep the experiment within reasonable boundaries.

As in the first experiment, we conducted additional regression analyses using the similarity category as the predictor and the mean similarity judgments as the response variable. An ANOVA comparing the quadratic and the linear model found no significant increase in fit, $F(1,7) = 0.47$, $p = .52$. The analysis of the linear model revealed a significant relationship between the similarity category and the mean similarity rating, $F(1,8) = 5.52$, $p = .047$ (intercept: 0.39, $95\%\ CI$ [0.28, 0.50], $t(8) = 8.48$, $p < .001$; category 0.02, $95\%\ CI$ [0.00, 0.04], $t(8) = 2.35$, $p = .047$). The model explained 40.8% of the variance ($R^2 = 0.408$). Again, the lack of a quadratic effect in this aggregated dataset compared to the first experiment likely stems from the diminished variance due to the smaller number of speaker pairs per category (10 vs. 500) and the reduced sample size (50 vs. 100). Despite these limitations, the findings suggest a consistent, incremental monotonic increase in similarity judgments for speaker pairs in higher similarity categories.

**Reliability and attenuation correction.** The test-retest reliability, indexed by the median of the individual Spearman Correlation between the first and the second similarity rating, was $Mdn\ r_s = 0.57$ ($Q_1 = 0.44$, $Q_3 = 0.65$), which can be considered as a fair test-retest reliability [80]. Since the cosine similarity values derived from the encoder are consistent, a single attenuation correction was performed to estimate the true correlation. Using the obtained reliability value yielded a correlation between the cosine values of the encoder and the participants' similarity ratings of $Mdn\ r_s = 0.48$ for the first experiment and $Mdn\ r_s = 0.31$ for the second experiment. Explanatory analyses showed a polynomial relationship between the first and the second rating (Fig 2). This indicates a stronger correlation for extreme (dis-)similarities.

The observed reliability is crucial in understanding the correlation between AI-generated cosine values and human similarity judgments. Any error or inconsistency in human judgments (as suggested by the reliability value less than 1) can attenuate or reduce the observed correlation. This means that the true correlation would likely be higher in the absence of such errors. Therefore, the obtained correlation between AI ratings and human judgments underestimates the actual strength of this relationship due to the influence of measurement error inherent in human judgments. Considering this reliability, our attenuation correction suggests that the true correlation is stronger than what is directly observed – even though the correlation values calculated by the attenuation correction are to be regarded as upper bounds.

The more pronounced correlations at extreme values of similarity, as indicated by the polynomial relationship, supports this view and demonstrates that people struggle to make reliable and consistent judgments for speaker pairs average in similarity. This finding does not contradict the outcomes of Experiment 1, where a quadratic relationship suggested difficulties in making nuanced ratings for more dissimilar speaker pairs. Moreover, the results added evidence to the notion that judging the similarity is inherently challenging, with more reliable assessments typically occurring at the extremes of the similarity spectrum.

**Consistency across raters.** Since we used a fixed set of stimuli for all participants, we also assessed the consistency of similarity judgments across different raters by calculating the Intraclass Correlation Coefficient (ICC) with the R package irr [81]. We employed a two-way model to evaluate the level of agreement on similarity judgments among the 44 raters across the 50 speaker pairs. Since participants rated each speaker pair twice, we analyzed only the

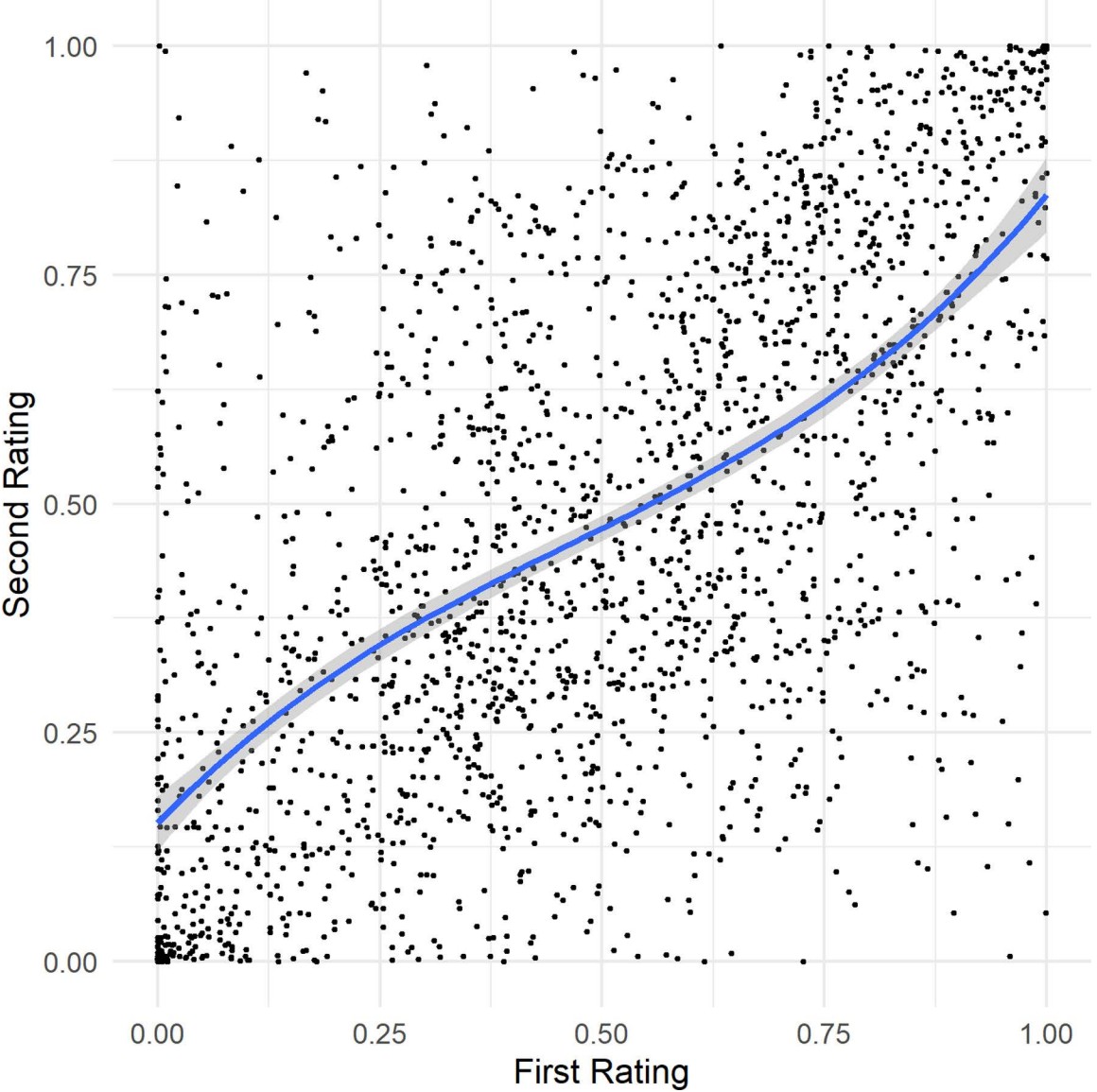

**Fig 2. Illustration of the results of Experiment 2.** The scatterplot shows the polynomial relationship between the first and the second similarity judgment and the 95% confidence interval of the regression line.

ratings from the first 50 trials. Where participants skipped the assessment of a speaker pair, we used the median of the other participants to replace the missing values – which was necessary in 17 of the 2200 cases. The results indicated a small to moderate level of agreement among the raters, ICC(A,1) = 0.31 (95% *CI* [0.23, 0.42], $F(49, 722)$ = 25.8, $p < .001$). Albeit these results suggest a consistent assessment of similarity across raters within the context of our study, there are also significant individual differences in judging the similarity of speaker pairs highlighting the difficulty to evaluate the similarity of two voices.

**General observations.** Participants skipped, on average, $M$ = 2.30 trials ($SD$ = 1.79) and needed, on average, $Mdn$ = 26.21 minutes to complete the experiment.

The findings confirm that AI-derived cosine values are predictive of human similarity judgments, though their strength varies depending on dataset restrictions and individual

variability. Stronger correlations at extreme similarity values underscore participants' difficulties in making reliable judgments for speaker pairs of average similarity. This pattern complements the quadratic relationship observed in Experiment 1, where the dissimilarity of speaker pairs posed challenges for nuanced ratings. Overall, the results reinforce that similarity judgments are inherently challenging and prone to individual differences, particularly in moderate similarity categories.

## Similarity judgments in relation to the own voice

The first two experiments demonstrated the encoders' ability to predict similarity judgments when the voices are those of other people. In the third experiment, we investigated whether this holds true if one of the voices is one's own voice. Since the perception of one's own voice depends on whether we are speaking or just listening to an audio sample of our voice [82], we manipulated this as a between-subjects factor. In the internal group, we only presented an audio sample of a speaker and asked participants to compare this sample with their own (internal) voice. In the external group, we presented an audio sample and additionally a sample of the participant, which was recorded prior to the experiment.

### Method

**Participants.** To ensure statistical consistency across experiments and facilitate meaningful comparisons, we used the same sample size of 100 participants in Experiment 3 (and the remaining experiments) as in Experiment 1. This approach minimizes potential discrepancies arising from differences in statistical power. Therefore, we recruited 100 new German participants via prolific. Ten were excluded because they detected fewer than two control trials. The mean age was $M = 25.76$ ($SD = 6.81$). Fifty-one participants were female, two diverse, and three did not specify their sex. Recruitment occurred from August 14, 2021, to August 30, 2021. Participants received £5.00 for their participation in the study.

**Materials, stimuli and procedure.** In the first session, each participant recorded five sentences. These recordings were used to compute the feature vector of their voice. We then calculated the cosine similarities of the participants' voice embeddings with all speakers of the same gender in our dataset. In order to achieve the highest possible variance in cosine similarities, we assigned each raw cosine similarity value to one of ten similarity categories – where the category boundaries from the first two experiments were used. Ten speakers were randomly selected from each category, resulting in a total of 100 speakers. Since the cosine similarity values are approximately normally distributed, the extreme categories would be underrepresented otherwise. If there were not enough speakers in the more extreme category, a speaker was chosen from the category closer to the mean. We picked one audio sample from our dataset for each speaker, trimmed it to a maximum length of 5 s, and normalized the volume.

In the second session, after checking the audio setup, in 100 trials the participants were asked to rate the dissimilarity of the presented voice in comparison to their own voice. If they were assigned to the external representation group, on each trial, participants were randomly presented with one of their five audio recordings, followed by another person's audio sample – without them having been instructed to listen to their recordings beforehand. In the internal representation group, only the speaker from our dataset was presented. In both groups, we simply asked the subjects to rate the similarity of the other person's voice to their own voice and, therefore, did not mention that their own voice might have an internal representation.

After presenting three introductory trials, every 30th trial contained recordings of two different speakers, none of which came from the participant. Participants were asked to detect

## Results

We used mixed models with participants as random effects, the raw cosine values as the independent variable, and judged similarity as the dependent variable. We compared an intercept-only model, a linear model, and a quadratic model. Additionally, we included the between-subjects factor as an interaction term. Weighting using the AIC scores in Table 3 showed a linear relationship (Fig 3) between cosine similarity values and participants' ratings. The model explained 21% of the variance in similarity ratings (intercept: 0.34, *95% CI* [0.32, 0.37], $t(101.1) = 26.32$, $p < .001$; cosine: 0.15, *95% CI* [0.12, 0.18], $t(8853) = 9.80$, $p < .001$; $R_c^2 = 0.21$). Whe*t*her one's voice was externally presented or not had no significant effect. The median of the individual Spearman Correlation between cosine similarities and similarity ratings was *Mdn* $r_s$ = 0.11 ($Q_1$ = 0.01, $Q_3$ = 0.23). Performing an attenuation correction yielded a Spearman Correlation of *Mdn* $r_s$ = 0.15. This reflects moderate predictive power at the individual level. Participants skipped on average $M = 2.11$ trials ($SD = 2.56$) and needed, on average, *Mdn* = 25.07 minutes to complete the experiment.

We conducted additional regression analyses, employing the similarity category as the predictor and the average similarity judgments as the dependent variable. An ANOVA contrasting the quadratic with the linear model just missed the threshold of significance, $F(1,7) = 4.27$, $p = .078$. The analysis of the linear model revealed a significant effect of the similarity category on the mean similarity ratings, $F(1,8) = 194.1$, $p < .001$ (intercept: 0.34, *95% CI* [0.33, 0.343], $t(7) = 98.34$, $p < .001$; ca*t*egory: 0.009 (*95% CI* [0.007, 0.010], $t(7) = 13.93$, $p < .001$). This model accounted for a significan*t* variance in mean similarity ratings, as indicated by an $R^2 = 0.96$.

As we noticed an overall decrease in similarity ratings when participants compared voices to their own voice, we conducted post-hoc Tukey-Kramer tests to investigate differences across the three experiments. Significant differences emerged between the average slider responses: $M_2 - M_1$ = 0.03, $t = 5.49$, $p < .001$; $M_3 - M_1$ = -0.08, $t = -18.80$, $p < .001$; $M_3 - M_2 =$ -0.11, $t = -20.44$, $p < .001$. These findings suggest a consistent bias where participants are less likely to judge voices as similar to their own.

Experiment 3 showed the encoders' ability to partially predict similarity judgments even when one of the voices is one's own voice. Unlike Experiments 1 and 2, Experiment 3 revealed a linear relationship between raw cosine similarity values and judged similarity, likely reflecting a general bias against perceiving self-voice similarities. The lower similarity ratings may stem from a *Need for Uniqueness* [83], where participants hesitate to identify other voices as similar to their own. Alternatively, heightened familiarity with one's own voice may increase sensitivity to subtle differences, leading to underestimation of similarity. Surprisingly, whether participants compared the presented voices to an internal mental representation or an

**Table 3. Model selection table for Experiment 3.**

| Model name | Degrees of freedom | Log-Likelihood | AIC | ΔAIC | Weight |
|---|---|---|---|---|---|
| Linear model | 4 | 245.52 | −483.0 | – | 0.896 |
| Quadratic model | 5 | 244.35 | −478.7 | 4.33 | 0.103 |
| Linear model + group | 6 | 240.98 | −470.0 | 13.07 | 0.001 |
| Quadratic model + group | 8 | 239.00 | −462.0 | 21.06 | 0 |
| Intercept-only model | 3 | 201.08 | −396.2 | 86.87 | 0 |

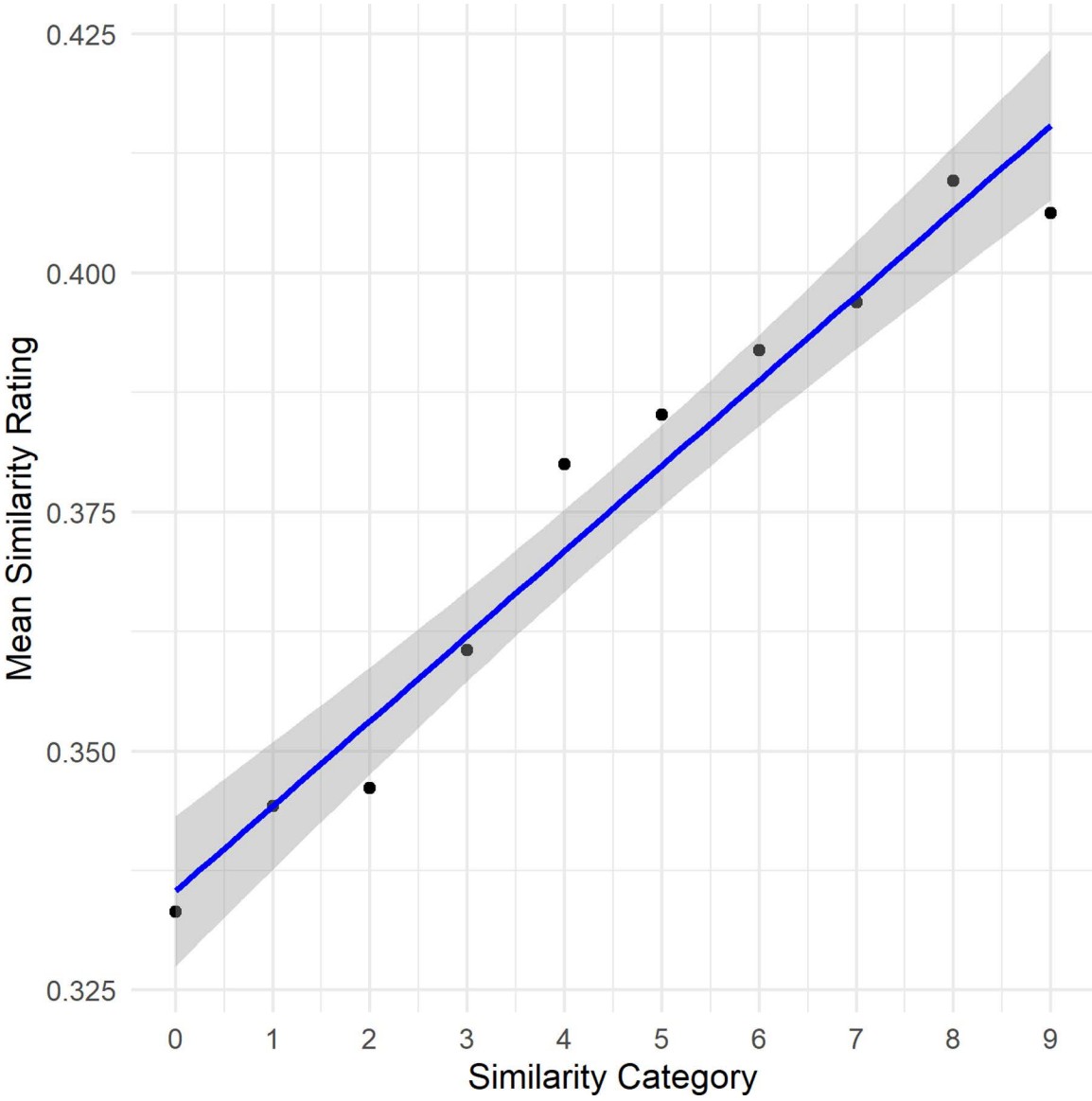

**Fig 3. Illustration of the results of Experiment 3.** Depicted is the quadratic relationship between cosine similarity categories and the participants' mean similarity ratings as well as the the 95% confidence interval of the regression line.

external recording of their own voice had no significant effect. This suggests that the internal representation of one's voice may serve as the dominant reference point.

## The beauty in average voices

Because the previous experiments indicated that cosine similarity is a valid proxy for perceived similarity, we investigated the cognitive consequences of voice similarity in the remaining experiments. In this experiment we focused particularly on the likability and trustworthiness of average voices. Our investigation was motivated by the concept of the beauty-in-averageness effect [39,40], which suggests that average features are often perceived as more attractive – even though they may not be optimally attractive [84].

This effect, well-documented in studies focusing on facial stimuli [85], may extend to auditory perceptions. By examining whether voices with average characteristics (determined by mean cosine similarity across our speaker dataset) are perceived as more likable and trustworthy, we wanted to explore whether this phenomenon transcends visual stimuli and applies to auditory perceptions as well.

## Method

**Participants.**  We recruited 100 new German participants via prolific. Two of them were excluded because they had less than 90 submitted ratings. The mean age was $M = 30.32$ ($SD = 11.64$). Forty-three participants were female, and two were diverse. Recruitment occurred from October 21, 2021, to October 26, 2021. Participants received £2.82 for their participation in the study.

**Materials, stimuli and procedure.**  We computed the cosine similarities of each male speaker with each other male speaker. We used the mean cosine similarity of a speaker as a measure of typicality. To obtain the highest possible variance in typicality, we assigned each mean cosine similarity value to one of ten similarity categories and randomly selected ten speakers from each category, resulting in a total of 100 speakers. We picked one audio sample from our dataset for each speaker, trimmed it to a maximum length of 5s, and normalized the volume. While the overall design of the experiment mirrored that of the second session in the third experiment – detecting sine tones to verify audio settings, performing three introductory trials, and judging 100 speakers – there were key differences. Instead of rating similarity, participants were asked to assess the likability and trustworthiness of the speakers using two continuous rating scales (ranging from "not at all" to "very"). Additionally, for the control trials, participants were required to detect audio samples from two female speakers.

## Results

Considering the AIC scores in Table 4, there was no significant effect of mean cosine similarity on likeability ratings (Fig 4). The model comparison for the trustworthiness ratings (see Table 5), revealed a quadratic relationship (Fig 5; intercept: 0.52, *95% CI* [0.49, 0.54], $t(178.19) = 39.35$, $p < .001$; mean cosine: -0.18, *95% CI* [-0.38, 0.02], $t(9627.08) =$ -1.88, $p = 0.06$; mean cosine$^2$: 0.74, *95% CI* [0.18, 1.30], $t(9627.08) = 2.57$, $p = .01$; $R_c^2 = 0.21$, indicating that the model explained 21% of the variance in trustworthiness ratings). The median of the individual Spearman Correlation between mean cosine similarities and trustworthiness ratings was $Mdn \ r_s = 0.04$ ($Q_1 = -0.03$, $Q_3 = 0.09$), reflecting weak associations. Participants skipped, on average, $M = 2.74$ trials ($SD = 1.12$) and needed, on average, $Mdn = 20.29$ minutes to complete the experiment.

Analyzing averaged data revealed no significant effects in mean cosine similarity values on likeability or trustworthiness ratings (all $p > .05$).

These results suggest that a voice's typicality (as captured by mean cosine similarity) does not affect likeability and has only a negligible influence on perceived trust. However, it

Table 4.  Model selection table for Experiment 4 – likeability.

| Model name | Degrees of freedom | Log-Likelihood | AIC | ΔAIC | Weight |
|---|---|---|---|---|---|
| Intercept-only model | 3 | 1231.85 | −2457.7 | – | 0.938 |
| Linear model | 4 | 1229.77 | −2451.5 | 6.18 | 0.043 |
| Quadratic model | 5 | 1229.95 | −2449.9 | 7.82 | 0.019 |

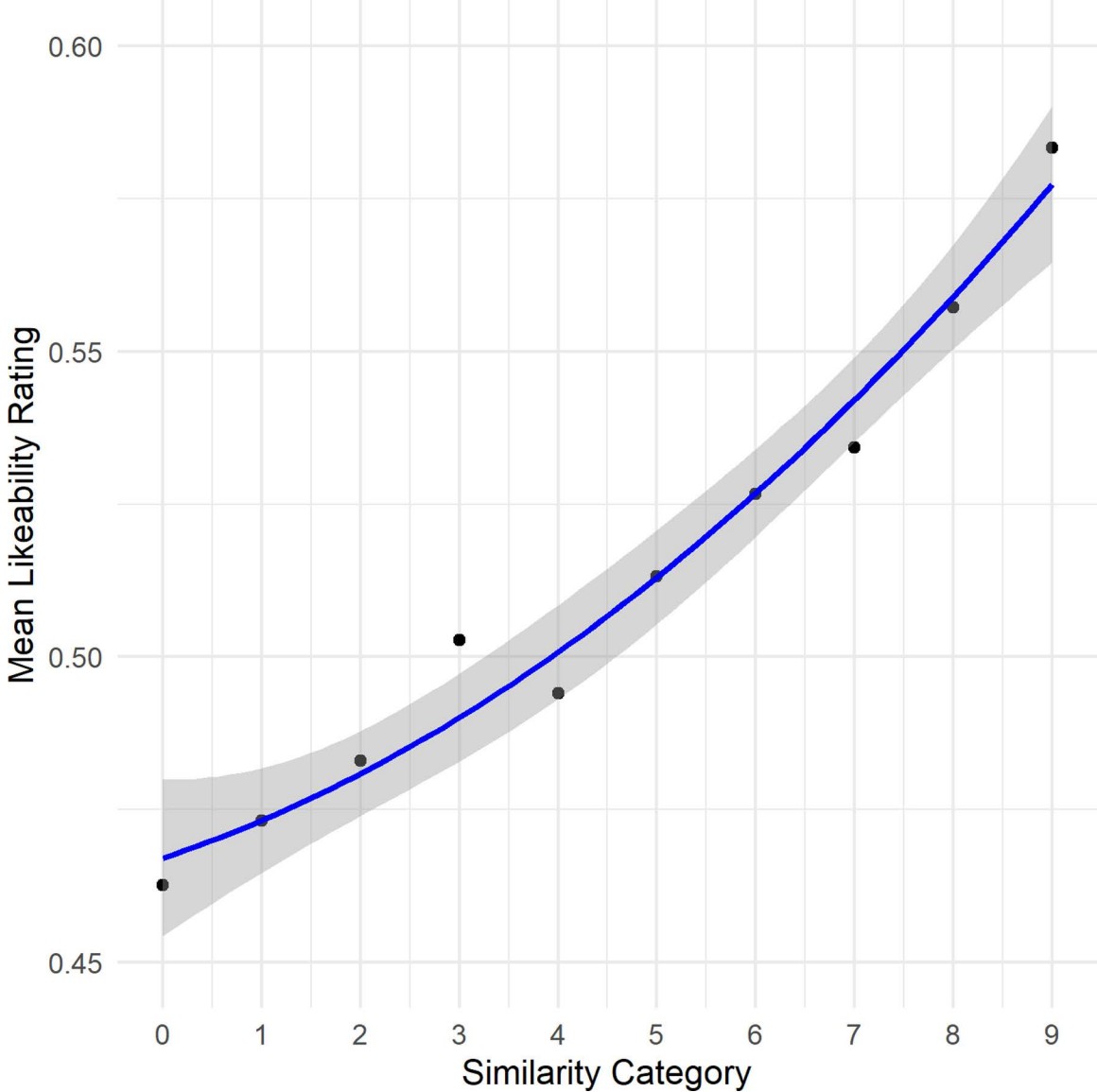

**Fig 4. First illustration of the results of Experiment 5.** Depicted is the quadratic relationship between cosine similarity categories and the participants' mean likeability ratings as well as the 95% confidence interval of the regression line.

**Table 5. Model selection table for Experiment 4 – trustworthiness.**

| Model name | Degrees of freedom | Log-Likelihood | AIC | ΔAIC | Weight |
|---|---|---|---|---|---|
| Quadratic model | 5 | 1162.81 | −2315.6 | – | 0.749 |
| Intercept-only model | 3 | 1159.19 | −2312.4 | 3.25 | 0.148 |
| Linear model | 4 | 1159.83 | −2311.7 | 3.96 | 0.104 |

should be noted that this could also be due to an insufficient number of particularly typical and atypical speakers or that the semantic content could have biased the evaluation. Previous research also pointed out that averageness has positive effects only in some dimensions but not in others [86].

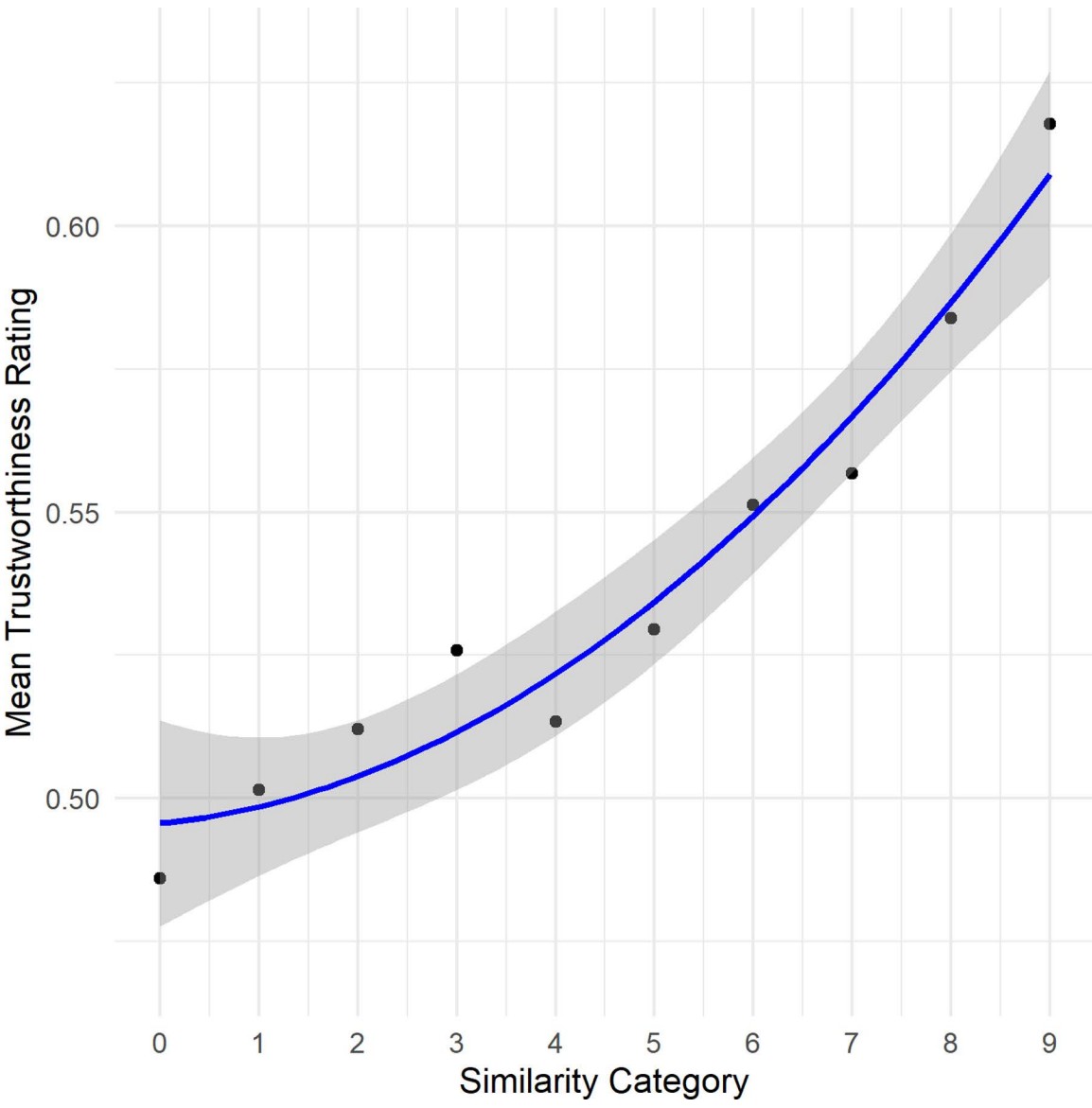

**Fig 5. Second illustration of the results of Experiment 5.** Depicted is the quadratic relationship between cosine similarity categories and the participants' mean trustworthiness ratings as well as the 95% confidence interval of the regression line.

## The attraction toward similar voices

In the final experiment, we investigated whether speakers with similar voices to one's own voice are perceived as more likable and trustworthy.

### Method

**Participants.** We recruited 100 new German participants via prolific. Seven were excluded because they had less than 90 submitted ratings. The mean age was $M = 31.04$ ($SD = 11.28$). Forty-five participants were female. Recruitment occurred from November 11, 2021, to November 29, 2021. Participants received £3.76 for their participation in the study.

**Materials, stimuli and procedure.**  As in the third experiment, the first session was used to record voice samples, compute the voiceprints, and sample 100 same-gender speakers from our dataset with a wide variety of cosine similarities. The second session was identical to Experiment 4.

## Results

We compared an intercept-only model, a linear model, and a quadratic model for both the likeability rating as well as the trustworthiness rating. Weighting using the AIC values in Table 6 showed a quadratic relationship between the voice similarity and likeability ratings (intercept: 0.47, *95% CI* [0.45, 0.49], $t(111) = 46.70$, $p < .001$; cosine: 0.11, *95% CI* [0.05, 0.18], $t(8863) = 3.55$, $p < 0.001$; mean cosine$^2$: 0.17, *95% CI* [0.04, 0.30], $t(8864) = 2.59$, $p = .009$; $R_c^2 = 0.19$, indicating that 19% of the variance in likeability ratings was explained by the model). The median of the individual Spearman Correlation between cosine similarities and likeability ratings was *Mdn* $r_s = 0.15$ ($Q_1 = 0.05$, $Q_3 = 0.25$), which, while modest, demonstrates a consistent positive relationship.

Weighting using the AIC values in Table 7 also showed a quadratic relation between the voice similarity and trustworthiness ratings (intercept: 0.50, *95% CI* [0.48, 0.52], $t(111.2) = 47.55$, $p < .001$; cosine: 0.06, *95% CI* [-0.01, 0.12], $t(8863) = 1.74$, $p = .08$; mean cosine$^2$: 0.31, *95% CI* [0.17, 0.45], $t(8865) = 4.45$, $p < .001$; $R_c^2 = 0.19$, indicating that 19% of the variance in likeability ratings was explained by the model). The median of the individual Spearman Correlation between cosine similarities and trustworthiness ratings was *Mdn* $r_s = 0.16$ ($Q_1 = 0.06$, $Q_3 = 0.25$), which, while once again modest, demonstrates a consistent positive relationship. Participants skipped, on average, $M = 5.95$ trials ($SD = 3.12$) and needed, on average, *Mdn* $= 19.06$ minutes to complete the experiment.

To investigate the relationship further, we employed the similarity category as the predictor, with the average likeability judgments serving as the dependent variable. The ANOVA comparison between the quadratic and the linear model demonstrated an improved fit for the quadratic model, $F(1,7) = 6.57$, $p = .04$. The quadratic model's analysis revealed a significant influence of the similarity category on the average likeability ratings, $F(2,7) = 134.5$, $p < .001$ (intercept: 0.467, *95% CI* [0.454, 0.480], $t(7) = 86.4$, $p < .001$; category at 0.005; *95% CI* [-0.001, 0.012], $t(7) = 1.916$, $p = .097$; category$^2$: 0.0008(*95% CI* [0.0001, 0.0015], $t(7) = 2.564$, $p = .04$). With $R^2 = 0.97$, this model accounted for a substantial portion of the variance.

The ANOVA comparison between the quadratic and linear models using the similarity categories as predictors and the corresponding mean trustworthiness ratings as the response

**Table 6.  Model selection table for Experiment 5 – likeability.**

| Model name | Degrees of freedom | Log-Likelihood | AIC | ΔAIC | Weight |
|---|---|---|---|---|---|
| Quadratic model | 5 | 1576.62 | −3143.2 | – | 0.635 |
| Linear model | 4 | 1575.07 | −3142.2 | 1.11 | 0.365 |
| Intercept-only model | 3 | 146638 | −2926.8 | 216.48 | 0 |

**Table 7.  Model selection table for Experiment 5 – trustworthiness.**

| Model name | Degrees of freedom | Log-Likelihood | AIC | ΔAIC | Weight |
|---|---|---|---|---|---|
| Quadratic model | 5 | 1206.21 | −2402.4 | – | 0.999 |
| Linear model | 4 | 1198.09 | −2388.2 | 14.25 | 0.001 |
| Intercept-only model | 3 | 1093.25 | −2180.5 | 221.92 | 0 |

variable indicated a more favorable fit for the quadratic model, $F(1,8) = 8.31$, $p = .02$. The quadratic model's analysis revealed a significant effect, $F(2,7) = 73.95$, $p < .001$ (intercept: 0.50, *95% CI* [0.48, 0.51], $t(7) = 65.141$, $p < .001$; ca*t*egory: 0.002, *95% CI* [-0.008, 0.011], $t(7) = 0.422$, $p = .686$; category$^2$: 0.001; *95% CI* [0.0002, 0.0022], $t(7) = 2.883$, $p = .02$). With $R^2 = 0.955$, *t*his model accounted for a significant proportion of the variance.

Taken together, these results demonstrate that the quadratic relationship between voice similarity and ratings of likeability and trustworthiness accounts for a substantial proportion of the variance. These findings suggest that while individual effect sizes are modest, the overall fit of the models underscores the practical relevance of voice similarity in shaping social perceptions. Indeed, these findings support the similarity-attraction hypothesis [44,45], in such that voices similar to one's own are perceived as more likable and trustworthy. The quadratic relationship suggests that the effect is stronger for higher levels of voice similarity. This effect likely stems from implicit egotism [47–49], suggesting that individuals evaluate self-associated traits positively or from social identity processes, in which perceived similarity fosters a sense of connection or group affiliation [55]. These results highlight the potential for AI systems to exploit similarity effects in personalized technologies, such as voice assistants, to influence user perceptions and behavior.

## Discussion

Speaker verification systems can compute numerical representations of human voices, so-called voiceprints. Whereas traditionally, speaker verification systems served, for example, as a forensic toolkit or as a biometric security feature in highly secured areas, the spread of deep learning technologies also increased the possibilities of utilizing voiceprints. Most importantly with regard to the present study, they could be used to design and shape the voice features of artificial voices. Despite the emerging importance of voiceprints in TTS systems, little research has been conducted on potential cognitive influences (including manipulations) of variations in the voice features of digital assistants. In the present set of experiments, we present such first evidence, including the methodological prerequisites for such an investigation.

The results of our first experiment indicated that the cosine similarity between voiceprints can predict voice similarity judgments (i.e., validity). The resulting quadratic relationship is likely due to disproportionate sensitivity to dissimilar voices. Since voices of close relatives are often similar and vary within a speaker depending on the time of day, physical condition, and context [87–89], the ability to discriminate between similar voices is a necessary skill for humans to learn. Below a certain threshold at which it is obvious that voices stem from different speakers, we are not aware of any reasonable explanation as to why it might be valuable to further differentiate between different levels of dissimilarity. Since the trained speaker verification system is agnostic regarding ecological advantages, it can differntiate even between dissimilar voices. Experiment 2 replicated these results. Moreover, the results revealed a relatively fair test-retest reliability of human similarity judgments. On the one hand, this implies that, in principle, correlations between similarity judgments and other cognitive variables should be observable. On the other hand, however, should such correlations arise, the numerical values most likely underestimate the true correlations since the reliability was far from being perfect.

The results of Experiment 3 revealed that the AI is also capable of partially predicting the perceived similarity between voices if one of the voices is one's own. Although the correlation was less pronounced, we found a linear relationship between cosine similarities and participant ratings. The most relevant difference for the experiment involving one's own voice relative to judging similarity between two unrelated voices (Experiments 1 and 2) was that

similarity ratings were generally lower. One possible explanation for this is people's *Need for Uniqueness* [83], which could make them more hesitant to classify a voice as similar to their own voice. Another interpretation could be that familiar voices are processed differently from unfamiliar voices [90,91]. The familiarity with one's own voice could thus increase the sensitivity for differences, which, in turn, might lead to an underestimation of the similarity. Interestingly, whether the participants were asked to compare a voice with their internal representation of their own voice, or an external recording of their voice had no substantial effect on the observed similarity judgments. This is surprising as one's own voice typically is not only transferred via air but also via bone. As there was no difference, we consider it likely that the internal representation of one's own voice may be the dominant reference point from which similarity judgments are made.

After establishing these essential methodological prerequisites for studying correlational relationships, we strived to investigate how voice features might affect basic cognitive evaluations. Following previous research on the beauty in averageness effect, we first investigated whether speakers are perceived as more likable and trustworthy if they have a more average voice. In contrast to other studies, we observed no evidence for a correlation between typicality and likeability and only a marginal effect of typicality on trustworthiness. However, there are substantial differences between previous studies and our experimental approach. Previous studies generated an average voice by creating composites, either by statistically averaging speakers [92] or by auditory morphing [93,94]. However, [95] consider that such composites lead to artifacts, especially an increased *harmonics-to-noise ratio* [96], which is discussed to decrease with age [97,98], in stressful situations [99], and cases of hoarseness [100]. Therefore, it is questionable whether the similarity or the more favorable change in the harmonics-to-noise ratio is the reason for the obtained results. In sum, we thus tend to consider that there is little evidence for a beauty-in-average effect of voices.

With regard to similarity to one's own voice, however, the results of our final experiment drew a different pattern of results. When using one's own voice as a reference point, similar voices are perceived as more likable and trustworthy. These findings match previous studies [101,102]. Again, however, the evidence from previous studies was rather weak, as they did not manipulate similarity directly [101] or just adjusted pitch (+/- 20Hz) or loudness (+/- 10 dB) [102]. Participants rated samples altered in loudness as more favorable compared to samples shifted in pitch. The authors concluded this pattern is due to the higher similarity of the loudness manipulated samples to the original recording. However, we are not convinced that participants perceive a recording of their voice as a recording from another person if it is just louder or quieter [103]. Therefore, recordings altered in pitch may not be compared to a similar voice but to one's own voice instead. Since people overestimate the attractiveness of their own voice, the results could be a consequence of this vocal implicit egotism [47]. It would have been much more consistent to manipulate the similarity by shifting the pitch by various degrees. Apart from that, shifting the pitch of a recording introduces far more noise than altering the loudness, making it more artificial and possibly unpleasant.

Our study's results underscore the potential for even brief voice recordings to be misused in shaping artificial voices in a way that influences people. The observed correlations between voiceprint similarity and human judgments of likability and trustworthiness (as seen in our final experiment) highlight a vulnerability in human perception. This could be exploited in TTS systems, where slight alterations of voice features aligned with a user's voiceprint could subtly sway their perceptions and behavior. Thus, while our research contributes to understanding the cognitive impact of voice similarity, it also opens discussions about ethical implications in the context of TTS technologies and voice assistants, where personalized voices might be used to manipulate user responses.

Even though the effect may have been relatively small in our experiments, due to the widespread use of voice assistants and the more elaborate methods available to large technology companies, the impact can be tremendous in absolute terms. This applies not only to the use of user-adapted voices to make interactions with the assistants more attractive but also to the impact on advertising messages and political propaganda.

## General limitations

While our study provides valuable insights into the relationship between voice similarity and cognitive evaluations, several limitations must be acknowledged.

A major limitation of this study arises from the wide range of voice similarity values that we were able to investigate. By including pairs of voices from a wide range of the similarity spectrum, we were able to ensure a solid understanding of the overall pattern; however, this broad spectrum may have diluted specific effects that are particularly pronounced in highly similar voices. Future studies should focus more on speakers with high cosine similarity values to better capture the nuances and practical implications of judgments in this critical range. This limitation also reflects a trade-off between experimental control and ecological validity. While our design provided valuable insights into general trends, focusing exclusively on highly similar voices may provide more precise and application-oriented results.

The use of open-source datasets, while providing a wide range of speaker voices, also introduced variability in audio quality, articulation, and linguistic content. These factors may not only have influenced participants' judgments but reduced the internal validity of the experiments. Future work should employ more controlled datasets or systematic manipulations of stimulus properties to reduce potential biases.

Our study emphasized static voice similarity judgments based on pre-recorded audio. Dynamic aspects of speech, such as conversational context, prosody, or situational factors, were not considered. These elements are likely to influence perceptions and warrant exploration in future research.

Finally, the online nature of the experiments presents challenges such as variable listening environments and participant compliance. Although attention checks and control trials were implemented, these measures cannot fully account for potential distractions or technical issues encountered by participants during the study.

## Conclusion

Our findings demonstrate that AI-derived cosine similarity measures effectively predict human voice similarity judgments and influence social evaluations. Across the first three experiments, we found significant relationships between the cosine similarity of voice embeddings and participants' similarity ratings, with a quadratic pattern emerging for judgments of other voices. This suggests that participants were particularly sensitive to highly similar and dissimilar voices, while intermediate similarity was more challenging to evaluate.

In Experiment 3, we extended this analysis to self-voice comparisons, revealing a general bias against perceiving other voices as similar to one's own voice. This bias likely stems from increased sensitivity to subtle differences in one's own voice or a Need for Uniqueness [83]. Comparisons to an internal mental representation or external audio recordings of the own voice yielded similar results, suggesting that the internal representation serves as a dominant reference.

Experiments 4 and 5 explored how voice similarity influences social perceptions such as likability and trustworthiness. Contrary to the beauty-in-averageness effect found in visual

stimuli, we found no evidence that average voices were perceived as more likable and only weak effects on trustworthiness. However, voices similar to one's own were judged as both more likable and trustworthy, supporting the similarity-attraction hypothesis and the influence of implicit egotism.

Our results highlight the potential of AI-generated cosine similarity as a tool for understanding voice perception. While individual effects were modest, the consistency of the findings underscores their practical relevance for voice-based technologies like personalized voice assistants or synthetic speech systems. Future research should focus on refining models for highly similar voices, exploring cross-linguistic generalizability, and addressing the ethical implications of voice similarity manipulations. This study advances our understanding of voice similarity's role in cognition and social interaction by bridging human perception and AI-driven voice representations.

## Author contributions

**Conceptualization:** Oliver Jaggy, Stephan Schwan, Hauke S. Meyerhoff.

**Data curation:** Oliver Jaggy.

**Formal analysis:** Oliver Jaggy.

**Investigation:** Oliver Jaggy.

**Methodology:** Oliver Jaggy.

**Project administration:** Oliver Jaggy.

**Software:** Oliver Jaggy.

**Supervision:** Stephan Schwan, Hauke S. Meyerhoff.

**Validation:** Oliver Jaggy.

**Visualization:** Oliver Jaggy.

**Writing – original draft:** Oliver Jaggy.

**Writing – review & editing:** Stephan Schwan, Hauke S. Meyerhoff.

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
