## [Decision Letter · Decision Letter 0]

1 Jul 2024

PONE-D-24-18772Do not trust your ears: Ai-determined similarity increases likability and trustworthiness of human voicesPLOS ONE

Dear Dr. Jaggy,

Thank you for submitting your manuscript to PLOS ONE. After careful consideration, we feel that it has merit but does not fully meet PLOS ONE’s publication criteria as it currently stands. Therefore, we invite you to submit a revised version of the manuscript that addresses the points raised during the review process.

We look forward to receiving your revised manuscript.

Kind regards,

Ying Shen, Ph.D.

Academic Editor

PLOS ONE

Journal Requirements:

Reviewers' comments:

Reviewer's Responses to Questions

**Comments to the Author**

1. Is the manuscript technically sound, and do the data support the conclusions?

Reviewer #1: Partly

Reviewer #2: Yes

Reviewer #3: Partly

Reviewer #4: Partly

2. Has the statistical analysis been performed appropriately and rigorously? 

Reviewer #1: Yes

Reviewer #2: Yes

Reviewer #3: I Don't Know

Reviewer #4: Yes

3. Have the authors made all data underlying the findings in their manuscript fully available?

Reviewer #1: Yes

Reviewer #2: Yes

Reviewer #3: Yes

Reviewer #4: Yes

4. Is the manuscript presented in an intelligible fashion and written in standard English?

Reviewer #1: Yes

Reviewer #2: Yes

Reviewer #3: Yes

Reviewer #4: Yes

5. Review Comments to the Author

Reviewer #1: Dear Sir/Madam,

Kindly receive my comments on the on the manuscript entitled:

“Do not trust your ears: Ai-determined similarity increases likability and trustworthiness

of human voices”

- I think the title is attractive, but it needs, from my perspective, to be more scientific than popular. I suggest deleting the first part, “Do not trust your ears.”

- The abstract lacks information on the approach and methods used.

- Line 60-63. … “Like fingerprints, the human voice can be used

61 to distinguish individuals from one another with a high degree of accuracy and gives insights

62 into the speaker's emotions and physical attributes”. The first part needs a reference.

- Line 131. ..”Only one study has explored the relationship between voice similarity estimates by humans and an automatic speaker recognition system [37]. I think this is a very strong statement. You can extend your search to cover the following study:

Grágeda N., Busso C., Alvarado E., García R., Mahu R., Huenupan F., Yoma N.B. Speech emotion recognition in real static and dynamic human-robot interaction scenarios (2025) Computer Speech and Language, 89, art. no. 101666, Cited 0 times.

DOI: 10.1016/j.csl.2024.101666

Rágeda N., Busso C., Alvarado E., Mahu R., Yoma N.B. Distant speech emotion recognition in an indoor human-robot interaction scenario (2023), 2023-August, pp. 3657 - 3661

DOI: 10.21437/Interspeech.2023-1169

- I suggest having the three research questions together as a list. This makes it clearer to the reader and more organized!

- Since “AI” is listed as a keyword and in the title, It is very necessary to have a paragraph about AI in general and then how it affects the voice and generates sounding voices and different platforms used.

- It is very important to explain the perspective of how the authors understand/perceive the terms “likability and trustworthiness”. A comprehensive argumentation should be followed with proper references.

- The method part should be extended. The authors need to explain more about all the proceeds of experiments 1 and 2, 3, 4, and 5. The sample also needs to be explained more (selection criteria, exceptions, exclusions, time of the experiment, is the gender balance considered?, How did you approach the participants??? Did you inform them about the whole process? Did you provide a trial session for them before the actual experiments, or was it spontaneous??

- I understand you have five experiments, each with a part for each method and result. However, this might be very confusing to the reader. I recommend you combine all methods in one part called “Materials and Methods”. Then, all results could be in one place under a title called “Results” but divided into parts as per experiments.

- Line 567: To me, this part is more discussion than a “general discussion.” If you want to keep it general, the logic says that you need to have another title called further/deep/focused discussion. Personally, I recommend having one title called “Discussion”.

I recommend the following articles to be listed under the general text about “AI”:

10.3390/buildings14030786

10.3390/buildings14030781

Reviewer #2: I would like to thank the authors for the innovative contribution to understanding the correlation between voiceprints and human perception. Using similarity assessments and d-vectors to generate speech that resembles a target speaker's voice, the authors concluded that cosine similarity is a valid measure for perceived similarity in this particular area with implications for cognitive research.

In my opinion, the significance of the study is clearly stated and the manuscript is well-written although some sections need to be re-drafted. For instance, there is a lack of a "Background/Related Work" section which could better describe the state-of-the-art in the field under study. The Introduction section provides some hints but I would rather suggest to create a section that could describe the existing related work more comprehensively. The General Discussion could be slightly expanded with more implications for practice in terms of Text-to-Speech systems, while the Conclusion section is quite short and must be expanded accordingly.

Regarding the methodological issues, the recruitment occurred between March 2021 (Experiment 1) and November 2021 (Experiment 5), which means that the data presented here has more than 2.5 years already when considering the last experiment conducted. The authors should also provide more details about the recruitment process on Prolific (e.g., total amount of USD paid per HIT). A better description of the interface and tasks presented to the participants would be helpful.

Minor issues:

The first part of the title needs a question mark (?)

I usually have doubts when authors claim that there is only one study addressing a certain aspect (e.g., "only one study has explored the relationship between voice similarity estimates by humans and an automatic speaker recognition system [37]").

Fig. 2 is labelled as showing an "Illustration of the results of Experiment 3" but it provides insights for the Experiment 2

Instead of naming sections as just "Experiment 1…2…3…", I would kindly suggest to create section titles with the main purpose of each experiment

Acronyms like LSTM, AIC, and AI should be described in the first time they are mentioned in the text

via prolific -> via Prolific (insert a link to the crowdsourcing platform as footnote)

MOSNet and MOSNET -> MOSNet (uniformity)

"[…] aggregated dataset compared to the first Experiment" -> "[…] aggregated dataset compared to the first experiment"

Reviewer #3: The present manuscript aims to explore the utility of voice similarity as measured by a neural network as a proxy for human perception of voice similarity as well as how this similarity might then guide perceptions of person likeability and trustworthiness. Across 5 experiments, the manuscript presents some evidence that neural network similarity measures could approximate human similarity measures to an extent when comparing unfamiliar voices (Exp. 1-2) or one’s own voice to unfamiliar voices (Exp. 3). There is also some indication that voices considered more typical (as measured by a neural network) are perceived somewhat more trustworthy (Exp.4) and that voices measured to be more similar to one’s own voice are also perceived as more likeable and trustworthy (Exp. 5). This work undoubtedly asks some interesting and very timely questions but I do have some concerns about the strength and generalisability of the reported effects. Please find my major and more minor comments detailed below.

- A large part of the Introduction is overly technical which could be particularly challenging for a broader audience. Personally, I am a behavioural scientist (not a computer scientist or a linguist) and I could not make sense of all the terminology used throughout. Also, more clarity is needed when reporting previous findings – most of the literature presented here was based on computer-based estimations, not behavioural ones but this is not made explicitly clear in the text. For example, at the start, it is stated that human voices can be used to distinguish individuals from one another – this is actually a very challenging and error-prone task for human observers (who have even been shown to struggle recognising their own voice). Same is true for the examples provided for recognising stress, emotions, demographics, etc. – they are mostly based on computer measurements. While human observers can also detect these in other’s voices, it is worth highlighting the perspective the authors are taking here to avoid any confusion.

- The overview presented at the end of the Introduction lacks clarity and does not help to get a good idea of the overall structure of this work. For example, it is not stated what the differences between Experiments 1-3 are. It might be clearer to simply describe each of the 5 experiments briefly. The authors provide no clear hypotheses and there is no justification as to why they are expecting quadratic relationships. Was this based on some previous research where such quadratic relationships were reported?

- There are some methodological clarifications needed throughout:

- What online platform was used to present these studies?

- Is there any particular reason why only male voices were used in Exp. 1?

- From my reading of the text, it seems like each pair of voices was rated by 2 participants only – this might not therefore lead to a very reliable, stable and generalisable behavioural similarity index.

- Were participants able to replay the pairs of voices?

- How was the unmarked continuous rating scale used to provide a numerical similarity measure?

- A power analysis is provided for Exp.2 but none of the others. It is also worth specifying the exact effect size used for this calculation. Is this the authors’ estimate of the relationship between first and second ratings or the relationship between AI-measured and behavioural voice similarity?

- In Exp.2, participants heard the voice pairs a second time but in the same order they first heard them in. The authors argue that this is to avoid variance from individual randomization but I’m concerned that this approach might artificially increase the correlation between first and second ratings, especially if participants can recall the pattern of responses from the first rating.

- In Exp.3 – were participants asked to indicate to what extent the recordings of their own voice sounded similar to their representation of their own voice?

- As stimuli were gender-matched in Exp.3, does that mean that participants who identified as diverse or did not specify their sex, were excluded?

- Were participants explicitly instructed that they will be hearing a recording of their own voice on each trial?

- How many trials did participants complete in Exp.3?

- Some further clarification is needed about the attention check used – it is stated that participants were asked to detect trials where voices were from two different speakers. Was that not true in all trials? My understanding is that voices from two speakers were presented at every trial.

- How many speakers were used in Exp.4?

- The authors argue that average faces are perceived as more attractive but it might be worth pointing out that this is not always the case. For example, Perrett et al. (1994, Nature) present some evidence against the averageness hypothesis.

- Looking at the data made available by the authors, it seems like some pairs of voices have a negative cosine similarity – this might be worth mentioning in the text together with how to interpret these negative values.

- How was the median correlation calculated?

- The main effect reported in the paper – the relationship between human- and AI-based measures of voice similarity seems rather inconsistent in size. In fact, the median correlations reported decrease with every next experiment.

- I also worry about the seemingly low within-rater consistency – it might indicate that people are not able to do this reliably enough. What is more, nothing is said about the linguistic content of the recordings. I’m assuming that different speakers had different utterances – therefore it might be that perceivers remember the utterance which can then artificially increase their consistency.

- I do not think it is appropriate to apply an attenuation correction due to the low perceiver reliability – low reliability means that perceivers might not be able to do this task with high levels of consistency, not that the correspondence between human and AI similarity measures might be overestimated.

- I realise that this might be a personal preference, but I would not advise describing an effect with a p value of .078 as marginally significant.

- In the Discussion, authors argue that a potential problem with voice averages (composites) is that they might be representative across age. Is their suggested alternative stable across time?

Reviewer #4: The authors present a series of experiments in which they test whether an AI AI-based voice similarity measure (voiceprint) corresponds to human similarity measures and whether voice similarity influences the voice’s trustworthiness and likeability. They report a correspondence between the voiceprint and human ratings as well as (in one experiment) an increase in perceived trustworthiness and likeability with similarity to one’s own voice. The topic is interesting and timely, and the experiments are well conducted and analyzed. While I applaud the authors for this, I see some serious flaws in the paper which can partly easily be mended, but partly would need at least a very thorough rethinking of the argument. It is not impossible that they are due to the fact that I am not an expert in the topic. But even then, I think they show that the authors are not making themselves sufficiently clear to non-experts. In fact, I accepted the review because I assumed trust would be a central theme; here I would have been an expert. The first part of the title misled me (and the abstract did not correct this). So, I would suggest a different title – and one that is less loud, by the way.

The authors claim relevance because of the introduction of a new methodological tool and the possibly important results and speak of a broad appeal of their results. This is not implausible, but this part specifically is not well developed. I would expect more precise arguments (or less strong claims).

The introduction is disorganized; the reader must construct the line of reasoning himself, understanding a series of scattered defined technical terms and finally figuring out what is actually the core of the later examination, the technical terms or human perception. Statements on the social relevance of the topic are thrown in, but not very integrated (and rather strongly formulated). It seems to me that the Introduction would have to be totally rewritten, and I am not sure that a tightly knit Introduction would lead to exactly the present experiments as an appropriate answer to the questions. I especially wonder whether the two parts – the test of the voiceprint and the test of the influence of similarity – need to be published together. Neither part is very strong, but putting them together in one paper does not make them stronger. Also, the research questions should be better developed and better derived from the introductory paragraphs.

I would make effect sizes and explained variance a much more prominent part of the presentation of results and the interpretation and conclusion. The relatively strong claims about the practical relevance of the study that are interspersed throughout the text only make sense if combined with a clear estimate and discussion of the practical relevance. What is a “fair amount” of explained variance in this context?

I wondered whether some of the more technical parts could be moved to a subsection in the Introduction. As such, it is one major source that distracts from a psychological argument in the Introduction.

For all experiments, there are long, dense, and often unexplained sections describing results and comparatively short sections discussing them. Methods should be better justified (sometimes explained), and the explanation what the results mean and how they are limited should be substantively expanded.

In several places it seems as if the authors – post hoc – explain away unexpected results (changes between experiments, low correlations) (e.g. 11 and 13). However, the respective objections are highly relevant for their study and its interpretation.

The General Discussion contains at least one argument that should have been considered pre-hoc instead of post-hoc (p. 23, threshold an ecological validity).

So, considering PLOS ONE’s criteria for Reviewing papers, there are (partly substantial) weaknesses in

• stating the main claims of the paper and showing how significant are they for the discipline,

• in supporting the claims with the data, or, more precisely, in discussing the amount of support in a balanced way

• in accessibility for non-specialists and organization of the paper

As the study shows much potential, a thoroughly revised version (maybe including new experiments) could be appropriate for PLOSE ONE.

Minors

The Introduction contains a sentence that includes “Forecasts predict that by 2024 …”. As we are already well into 2024, it would be appropriate to compare the forecast to reality here.

p. 7: It is unclear to me, how a research question can extend findings.

p. 9: Here, a slider is introduced suddenly. Could be explained better in the methods section.

Experiment sections should not end with new (secondary) information (p. 15).

6. PLOS authors have the option to publish the peer review history of their article (what does this mean? ). If published, this will include your full peer review and any attached files.

**Do you want your identity to be public for this peer review?** For information about this choice, including consent withdrawal, please see our Privacy Policy .

Reviewer #1: No

Reviewer #2: No

Reviewer #3: No

Reviewer #4: No

---

## [Author Response · Author response to Decision Letter 1]

31 Dec 2024

12.17.2024

Revision of Manuscript PONE-D-24-18772

Dear Ying Shen, Ph.D.,

We write regarding our manuscript, “Ai-determined similarity increases likability and trustworthiness of human voices” (PONE-D-24-18772), for which you have served as Editor at PLOS ONE.

Based on for reviews, you requested that we address the comments of the reviewers. We have done so through the detailed responses included in this document (please see below) and appropriate revisions to our manuscript. We hope you will agree that our paper has benefited from these revisions. We look forward to hearing from you about whether this work will now constitute a package that is acceptable for publication in PLOS ONE.

Please do not hesitate to contact us if you have any further questions and thank you for your consideration!

Sincerely,

Oliver Jaggy, Stephan Schwan & Hauke S. Meyerhoff

Responses to Reviewer #1

Kindly receive my comments on the on the manuscript entitled: “Do not trust your ears: Ai-determined similarity increases likability and trustworthiness of human voices” I think the title is attractive, but it needs, from my perspective, to be more scientific than popular. I suggest deleting the first part, “Do not trust your ears.”

That's true. The title has been changed.

The abstract lacks information on the approach and methods used.

Thank you for this comment. We have revised the abstract as follows:

“Modern artificial intelligence (AI) technology is capable of generating human sounding voices that could be used to deceive recipients in various contexts (e.g., deep fakes). Given the increasing accessibility of this technology and its potential societal implications, the present study conducted online experiments using original data to investigate the validity of AI-based voice similarity measures and their impact on trustworthiness and likability. Correlation analyses revealed that voiceprints – numerical representations of voices derived from a speaker verification system – can be used to approximate human (dis)similarity ratings. With regard to cognitive evaluations, we observed that voices similar to one’s own voice increased trustworthiness and likability, whereas average voices did not elicit such effects. These findings suggest a preference for self-similar voices and underscore the risks associated with the misuse of AI in generating persuasive artificial voices from brief voice samples.” (p. 2)

Line 60-63. … “Like fingerprints, the human voice can be used

61 to distinguish individuals from one another with a high degree of accuracy and gives insights

62 into the speaker's emotions and physical attributes”. The first part needs a reference.

We included the following references:

1. Doddington GR. Speaker Recognition—Identifying People by Their Voices. Proceedings of the IEEE. 1985;73: 1651–1664. doi:10.1109/PROC.1985.13345

2. Li H, Xu C, Rathore AS, Li Z, Zhang H, Song C, et al. VocalPrint: exploring a resilient and secure voice authentication via mmWave biometric interrogation. Proceedings of the 18th Conference on Embedded Networked Sensor Systems. Virtual Event Japan: ACM; 2020. pp. 312–325. doi:10.1145/3384419.3430779

Line 131. ..”Only one study has explored the relationship between voice similarity estimates by humans and an automatic speaker recognition system [37]. I think this is a very strong statement. You can extend your search to cover the following study:

Grágeda N., Busso C., Alvarado E., García R., Mahu R., Huenupan F., Yoma N.B. Speech emotion recognition in real static and dynamic human-robot interaction scenarios (2025) Computer Speech and Language, 89, art. no. 101666, Cited 0 times.

DOI: 10.1016/j.csl.2024.101666

Rágeda N., Busso C., Alvarado E., Mahu R., Yoma N.B. Distant speech emotion recognition in an indoor human-robot interaction scenario (2023), 2023-August, pp. 3657 - 3661

DOI: 10.21437/Interspeech.2023-1169

We appreciate the references. However, since they do not deal with the comparison of human and AI similarity assessment of voices, we have not cited them in the suggested place. However, as they investigate emotion perception, we have mentioned them in the introduction. Additionally, we added the following qualification:

“[…] to the best of our knowledge, there is only one study that has investigated the relationship between voice similarity estimates by humans and an automatic speaker recognition system [39].” (p. 5)

I suggest having the three research questions together as a list. This makes it clearer to the reader and more organized!

We included the following listing at the end of our introduction (p. 9):

Based on the above considerations, we investigated

whether the cosine similarity derived from the trained neural network correlates with human similarity judgments (Exp 1-3).

whether speakers with prototypical voices are judged as more likeable and trustworthy (Exp 4).

whether speakers with similar voiceprints to the corresponding participants are perceived as more likable and trustworthy (Exp 5).

Since “AI” is listed as a keyword and in the title, It is very necessary to have a paragraph about AI in general and then how it affects the voice and generates sounding voices and different platforms used.

Thank you for your advice. We have insert a brief paragraph at the beginning of the revised introduction.

“Artificial intelligence (AI) has become integral to modern life and is revolutionizing how people interact with technology and process information. From autonomous vehicles to personalized recommendation systems, AI's ability to analyze and replicate human-like behaviors profoundly impacts all industries. One particularly relevant application is in the field of speech technology, in which AI systems not only recognize and synthesize speech but also simulate individual voice characteristics. This capability opens new avenues for personalized interactions, such as matching voice assistants to a user's voice profile or augmenting that profile, illustrating the interplay between technology, identity, and human perception.” (p. 3)

It is very important to explain the perspective of how the authors understand/perceive the terms “likability and trustworthiness”. A comprehensive argumentation should be followed with proper references.

Thank you. We have restructured the introduction and introduced a section on likeability and trustworthiness:

“Likeability and trustworthiness are foundational attributes that significantly influence social interactions and relationships. Research has demonstrated that individuals judged more likable by others are more persuasive, often receiving preferential treatment and social support [27–30], and that similar others are also perceived as more likeable [31]. Similarly, trustworthiness is central in fostering long-term (business) relationships and ensuring effective collaboration, as it mitigates uncertainty, reduces the perceived risk in interactions and increases predictability [32–36]. From an evolutionary perspective, trustworthiness likely signals an individual’s reliability and cooperative intent, essential for fostering social cohesion and reciprocal behaviors within groups. Similarly, likeability facilitates social bonding by eliciting positive affect and reducing interpersonal tension, enhancing collaboration and mutual support. Thus, since these constructs are integral to social evaluation processes, the use of likability and trustworthiness as dependent variables is critical to understanding the impact of voice similarity.” (p.6)

The method part should be extended. The authors need to explain more about all the proceeds of experiments 1 and 2, 3, 4, and 5.

The sample also needs to be explained more (selection criteria, exceptions, exclusions, time of the experiment, is the gender balance considered?

As mentioned in the text, for each experiment, we recruited new German participants, with nationality being our only selection criterion.

Regarding the "time of the experiment," if this refers to the duration, the average length of each experiment is provided in the respective results section. If it refers to the actual time of day, the experiments typically occurred during daytime hours, although we did not have complete control over this aspect. Additionally, it is unclear how this information would enhance the readers' understanding, but we are open to providing further clarification if needed.

Regarding gender balance, we ensured equal representation in our first experiment. As no gender-related effects were observed, gender balance was not a concern in the subsequent experiments:

“Since previous research found evidence for an own-gender bias in the ability of voice identification [73] and gender differences in voice processing [74,75], we included participants gender as an additional factor. […] Numerically, female participants rated the similarity slightly higher; however, the inclusion of gender as an additional factor is not justified given the AIC values.” (p.13)

How did you approach the participants???

As noted, we used Prolific (i.e., an online platform in which participants complete experiments) to recruit participants, meaning we did not engage with them directly or in a personalized manner.

Did you inform them about the whole process? Did you provide a trial session for them before the actual experiments, or was it spontaneous??

Although we briefly mentioned the use of introductory trials, we have clarified and emphasized this more explicitly article. In the methods section of Experiment 3, we included the following additional details:

“After presenting three introductory trials, every 30th trial contained recordings of two different speakers, none of which came from the participant.” (p. 22)

In the methods section of Experiment 4, we included the following additional details:

“While the overall design of the experiment mirrored that of the second session in the third experiment —detecting sine tones to verify audio settings, performing three introductory trials, and judging 100 speakers — there were key differences.” (p.25)

I understand you have five experiments, each with a part for each method and result. However, this might be very confusing to the reader. I recommend you combine all methods in one part called “Materials and Methods”. Then, all results could be in one place under a title called “Results” but divided into parts as per experiments.

This is, of course, a valid suggestion and has already been considered by us. As the individual experiments build on each other and we believe that this makes the individual experiments easier to understand, we prefer the chosen structure. However, we have tried to emphasize differences between the methods.

Line 567: To me, this part is more discussion than a “general discussion.” If you want to keep it general, the logic says that you need to have another title called further/deep/focused discussion. Personally, I recommend having one title called “Discussion”.

Good point, we renamed it to discussion.

I recommend the following articles to be listed under the general text about “AI”:

10.3390/buildings14030786

10.3390/buildings14030781

We could not see how the papers mentioned are related to our manuscript. Even though we have added them to our list of articles to be considered in the future, we have decided to cite more broadly due to the lack of a current connection.

Responses to Reviewer #2

I would like to thank the authors for the innovative contribution to understanding the correlation between voiceprints and human perception. Using similarity assessments and d-vectors to generate speech that resembles a target speaker's voice, the authors concluded that cosine similarity is a valid measure for perceived similarity in this particular area with implications for cognitive research.

Many thanks for these words of encouragement.

In my opinion, the significance of the study is clearly stated and the manuscript is well-written although some sections need to be re-drafted. For instance, there is a lack of a "Background/Related Work" section which could better describe the state-of-the-art in the field under study. The Introduction section provides some hints but I would rather suggest to create a section that could describe the existing related work more comprehensively.

Thank you for pointing this out, we have restructured and partially rewritten the introduction and added, for example, information about similarity attraction effects (pp. 7-9).

The General Discussion could be slightly expanded with more implications for practice in terms of Text-to-Speech systems, while the Conclusion section is quite short and must be expanded accordingly.

Thank you very much for this suggestion. Throughout the manuscript we added information what the results might imply from a practical point of view. We also added a general limitations section and rewrote the conclusion section as follows:

“Our findings demonstrate that AI-derived cosine similarity measures effectively predict human voice similarity judgments and influence social evaluations. Across the first three experiments, we found significant relationships between the cosine similarity of voice embeddings and participants' similarity ratings, with a quadratic pattern emerging for judgments of other voices. This suggests that participants were particularly sensitive to highly similar and dissimilar voices, while intermediate similarity was more challenging to evaluate.

In Experiment 3, we extended this analysis to self-voice comparisons, revealing a general bias against perceiving other voices as similar to one's own voice. This bias likely stems from increased sensitivity to subtle differences in one's own voice or a Need for Uniqueness [88]. Comparisons to an internal mental representation or external audio recordings of the own voice yielded similar results, suggesting that the internal representation serves as a dominant reference.

Experiments 4 and 5 explored how voice similarity influences social perceptions such as likability and trustworthiness. Contrary to the beauty-in-averageness effect found in visual stimuli, we found no evidence that average voices were perceived as more likable and only weak effects on trustworthiness. However, voices similar to one's own were judged as both more likable and trustworthy, supporting the similarity-attraction hypothesis and the influence of implicit egotism.

Our results highlight the potential of AI-generated cosine similarity as a tool for understanding voice perception. While individual effects were modest, the consistency of the findings underscores their practical relevance for voice-based technologies like personalized voice assistants or synthetic speech systems. Future research should focus on refining models for highly similar voices, exploring cross-linguistic generalizability, and addressing the ethical implications of voice similarity manipulations. This study advances our understanding of voice similarity's role in cognition and social interaction by bridging human perception and AI-driven voice representations.” (pp. 33-34)

Regarding the methodological issues, the recruitment occurred between March 2021 (Experiment 1) and November 2021 (Experiment 5), which means that the data presented here has more than 2.5 years already when considering the last experiment conducted.

This is of course an appropriate observation. However, there seems to be no reason why we would expect our experiments not to replicate if we would rerun them today.

The authors should also provide more details about the recruitment process on Prolific (e.g., total amount of USD paid per HIT). A better description of the interface and tasks presented to the participants would be helpful.

We think this is a very good suggestion. We included information of the received payment in every participants section:

Experiment 1: “Participants received £3.45 for their participation in the study.”

Experiment 2: “Participants received £4.36 for their participation in the study.”

Experiment 3: “Participants received £5.00 for their participation in the study.”

Experiment 4: “Partic

---

## [Decision Letter · Decision Letter 1]

24 Jan 2025

Ai-determined similarity increases likability and trustworthiness of human voices

PONE-D-24-18772R1

Dear Oliver Jaggy,

We’re pleased to inform you that your manuscript has been judged scientifically suitable for publication and will be formally accepted for publication once it meets all outstanding technical requirements.

Kind regards,

Ying Shen, Ph.D.

Academic Editor

PLOS ONE

Additional Editor Comments (optional):

Reviewers' comments:

Reviewer's Responses to Questions

**Comments to the Author**

1. If the authors have adequately addressed your comments raised in a previous round of review and you feel that this manuscript is now acceptable for publication, you may indicate that here to bypass the “Comments to the Author” section, enter your conflict of interest statement in the “Confidential to Editor” section, and submit your "Accept" recommendation.

Reviewer #2: All comments have been addressed

Reviewer #4: All comments have been addressed

2. Is the manuscript technically sound, and do the data support the conclusions?

Reviewer #2: Yes

Reviewer #4: Yes

3. Has the statistical analysis been performed appropriately and rigorously? 

Reviewer #2: Yes

Reviewer #4: Yes

4. Have the authors made all data underlying the findings in their manuscript fully available?

Reviewer #2: Yes

Reviewer #4: Yes

5. Is the manuscript presented in an intelligible fashion and written in standard English?

Reviewer #2: Yes

Reviewer #4: Yes

6. Review Comments to the Author

Reviewer #2: The revised manuscript fully incorporates my comments, thank you for your work. Please proofread your English carefully.

Reviewer #4: I thank the authors for taking my comments seriously and addressing them. They have done so very thoroughly and convincingly. The paper is a fine contribution to oud knowledge and I wish them many interested readers.

7. PLOS authors have the option to publish the peer review history of their article (what does this mean? ). If published, this will include your full peer review and any attached files.

**Do you want your identity to be public for this peer review?** For information about this choice, including consent withdrawal, please see our Privacy Policy .

Reviewer #2: No

Reviewer #4: No

---

## [Editor Report · Acceptance letter]

PONE-D-24-18772R1

PLOS ONE

Dear Dr. Jaggy,

I'm pleased to inform you that your manuscript has been deemed suitable for publication in PLOS ONE. Congratulations! Your manuscript is now being handed over to our production team.

Kind regards,

on behalf of

Dr. Ying Shen

Academic Editor

PLOS ONE